# Efficient hybrid CNN-transformer model for accurate blood cancer detection

Wiem Abdelbaki[1], Muhammad John Abbas[2], Fathimathul Rajeena P. P[3], Inzamam Mashood Nasir[2,4], Deema Mohammed Alsekait[5], Adel Thaljaoui[6] and Diaa Salama AbdElminaam[7,8]

[1] College of Engineering and Technology, American University of the Middle East, Egaila, Kuwait
[2] Center of Real-World AI Research, Kaunas, Lithuania
[3] Computer Science Department, College of Computer Science and Information Technology, King Faisal University, Alhasa, Saudi Arabia
[4] Centre of Real Time Computer Systems, Kaunas University of Technology, Kaunas, Lithuania
[5] Department of Information Technology, College of Computer and Information Sciences, Princess Nourah bint Abdulrahman University, Riyadh, Saudi Arabia
[6] Department of Computer Science and Information College of Science Zulfi, Majmaah University Al-Majmaah, Saudi Arabia
[7] Jadara Research Center, Jadara University, Irbid, Jordan
[8] MEU Research Unit, Middle East University, Amman, Jordan

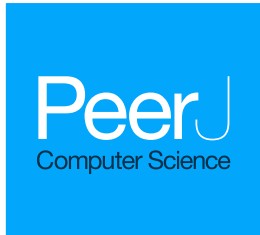

Corresponding authors
Deema Mohammed Alsekait,
dmalsekait@pnu.edu.sa
Adel Thaljaoui, adel.t@mu.edu.sa

## ABSTRACT

When the number of white blood cells (WBCs) in the human body becomes imbalanced, leukemia (blood cancer) can develop, affecting individuals of all ages. Early and accurate detection is crucial for improving patient outcomes, yet manual diagnosis is time-consuming and prone to subjectivity. This study proposes an efficient hybrid convolutional neural network (CNN)-transformer model for automated blood cancer detection, integrating convolutional layers for localized feature extraction with transformer-based attention mechanisms for capturing global dependencies. The architecture employs depthwise separable convolutions, efficient multi-head self-attention (EMHSA), and an efficient multilayer perceptron (EMLP) block, optimized *via* Bayesian hyperparameter tuning. The proposed model was evaluated on two publicly available datasets: the Blood Cancer dataset (binary classification) and the Blood Cells Cancer (ALL) dataset (four-class classification). Using a 50:50 training-testing split, the model achieved 100% accuracy, 100% precision, 100% recall, and 100% F1-score on the Blood Cancer dataset, and 99% accuracy, 99% precision, 100% recall, and 99% F1-score on the ALL dataset. Additional experiments with 80:20, 70:30, and 60:40 splits confirmed consistent performance above 98% across all metrics, demonstrating strong robustness. The model contains only 2.04 million trainable parameters, significantly fewer than standard CNN or transformer-based architectures, making it computationally lightweight and suitable for deployment in resource-constrained clinical environments. These results highlight the potential of the proposed hybrid framework to provide accurate and efficient blood cancer classification, advancing the applicability of deep learning in hematological diagnostics.

# INTRODUCTION

Stem cells located in the bone marrow play a vital role in the production and replenishment of blood cells. Human blood is composed of approximately 55% plasma and 45% cellular components, which include red blood cells responsible for oxygen transport and carbon dioxide removal, white blood cells that defend against infections, and platelets that regulate clotting and hemorrhage. Under normal physiological conditions, cells proliferate and older cells undergo apoptosis, thereby creating space for new cells. In malignancies such as leukemia, this balance is disrupted due to the absence of apoptosis in aged cells, resulting in an abnormal accumulation of immature and dysfunctional white blood cells. These leukemic cells weaken the immune response, leaving the body more susceptible to infections. Leukemia is diagnosed through blood tests and bone marrow biopsies, and it is classified into four major categories: acute lymphoblastic leukemia (ALL), acute myeloid leukemia (AML), chronic lymphocytic leukemia (CLL), and chronic myeloid leukemia (CML). Each type impacts the blood, bone marrow, and lymphatic system in different ways but shares the common feature of abnormal hematopoiesis.

Conventional diagnostic practices for leukemia often rely heavily on manual inspection of microscopic blood smear images. While effective, such human-supervised approaches are subject to variability, operator fatigue, and potential misdiagnosis. As medical imaging becomes increasingly central to hematological diagnostics, the demand for computer-aided diagnostic systems that minimize human error and maximize efficiency has grown. Machine learning and deep learning methods have shown significant promise in reducing operator dependency, automating routine diagnostic processes, and improving reproducibility (*Rawat et al., 2017*). Among the critical steps in this workflow, segmentation plays a vital role by isolating relevant structures such as leukocytes for further analysis, thereby directly impacting the accuracy of classification tasks (*Mohammed & Abdulla, 2021*).

Research in automated leukemia detection has explored a range of segmentation and classification techniques. Initial methods relied on handcrafted features, such as color, texture, and shape descriptors, which were subsequently used in combination with classical classifiers. Although these approaches demonstrated the potential for automation, their performance was constrained by their sensitivity to variations in staining, imaging conditions, and morphological heterogeneity. The emergence of machine learning, and more recently deep learning, has shifted the focus toward models capable of learning discriminative features directly from raw data, eliminating the need for extensive handcrafted preprocessing. This transition has been particularly important in medical imaging, where dataset limitations and variability remain challenging factors.

Convolutional neural networks (CNNs) have shown remarkable versatility across domains, particularly in medical imaging, agriculture, and security. In healthcare, they are applied to skin cancer (*Nasir et al., 2024a*, *2024b*), brain tumors (*Tehsin, Nasir & Damaševičius, 2024a*, *2024b*, *2025*), and breast cancer classification (*Nasir, Alrasheedi & Alreshidi, 2024*; *PP & Tehsin, 2025*), often enhanced with attention mechanisms and explainability methods like XGrad-CAM. In agriculture, CNN-based models support

wheat disease and head detection using ensemble and transformer-based approaches (*Yousafzai et al., 2025a*, *2025b*). Beyond healthcare, CNNs contribute to online news categorization, digital signature verification, and image encryption (*Yousafzai et al., 2024*; *Tehsin et al., 2024*; *Malik et al., 2024*), demonstrating their adaptability across varied tasks. Despite these advances, the choice of model architecture remains critical in balancing performance and efficiency. CNNs have become the backbone of medical image analysis due to their ability to capture local spatial features and hierarchical representations. However, CNNs exhibit limitations in modeling long-range dependencies, which are essential for capturing the broader context of blood smear images. On the other hand, transformer-based models excel in global context modeling but lack the inductive biases necessary for fine-grained morphological feature extraction, such as cell boundaries and internal textures. This dichotomy motivates the exploration of hybrid approaches that combine the complementary strengths of CNNs and transformers.

In this article, we propose an efficient hybrid CNN-transformer model designed specifically for blood cancer detection. The model integrates CNN blocks for localized feature extraction, transformer-based efficient multi-head self-attention (EMHSA) modules for capturing global correlations, and efficient multilayer perceptron (EMLP) blocks for parameter reduction without compromising representational capacity. With only 2.04 million parameters, the architecture achieves a lightweight footprint while maintaining high accuracy, outperforming conventional transformer models that often exceed 20 million parameters. This efficiency makes it well-suited for deployment in real-world, resource-constrained environments. The remainder of this article is structured as follows: 'Related Work' presents the proposed model in detail, 'Experiments' reports experimental results and discussion, and 'Conclusions' concludes with potential future directions.

The article is divided as follows: The literature review is presented in 'Efficient Hybrid CNN-Transformer Model'. The proposed model is fully detailed in 'Efficient Hybrid CNN-Transformer Model' while 'Experiments' covers experimental results and discussions. 'Conclusions' concludes this article with future directions.

## RELATED WORK

Research on automated leukemia detection has been extensive, spanning from traditional machine learning techniques to state-of-the-art deep learning models. Early efforts primarily relied on segmentation-based pipelines. A notable contribution in this category is the hybrid ellipse fitting (EF) approach proposed in *Das et al. (2021)*, which introduced accurate seed point detection by combining algebraic and geometric edge-finding methods, leading to improved segmentation performance. Similarly, *Shafique et al. (2019)* employed support vector machines (SVMs) using color and shape descriptors for the classification of ALL, demonstrating the potential of handcrafted features in clinical applications. Segmentation accuracy directly influenced classification outcomes, and therefore considerable emphasis was placed on extracting robust features from stained blood smear images.

Further refinements incorporated advanced feature extraction and dimensionality reduction methods. For example, gray-level co-occurrence matrix (GLCM) features have

been widely employed to capture texture patterns relevant to leukemia cells. In such workflows, probabilistic principal component analysis (PCA) was used to reduce the feature dimensionality, and Random Forest classifiers were applied to improve classification robustness (*Shafique et al., 2019*). Along the same lines, *Vogado et al. (2018)* designed a highly effective SVM-based classifier for ALL, while *Das & Meher (2021a)* proposed a PCA-driven reduction strategy combined with SVM for leukemia detection. More recently, *Mohammed & Abdulla (2021)* explored statistical, geometrical, and discrete cosine transform (DCT)-based features, coupled with SVM classification, achieving encouraging results for ALL analysis. Complementary approaches using K-nearest neighbor (KNN)-based classifiers were investigated by *Sahlol, Abdeldaim & Hassanien (2019)* and *Abdeldaim et al. (2018)*, who extracted color, texture, and shape features from the ALLIDB2 database, further highlighting the diversity of traditional methods applied to the problem.

Specific contributions have also targeted AML classification. *Agaian, Madhukar & Chronopoulos (2014)* proposed an SVM-based approach to distinguish AML from non-AML blood smear samples using the ASH database, highlighting the clinical applicability of machine learning in hematological malignancy detection. *Madhukar, Agaian & Chronopoulos (2012)* and *Kumar & Udwadia (2017)* followed a similar strategy with SVM classifiers using texture and geometric descriptors, also on the ASH database. These studies demonstrated the viability of traditional classifiers when paired with carefully crafted features, but their reliance on large amounts of domain-specific feature engineering limited scalability.

The limitations of handcrafted approaches led to the adoption of transfer learning and deep CNNs, which have proven effective for small medical datasets. *Das, Pradhan & Meher (2021)* demonstrated the effectiveness of transfer learning by modifying ResNet50 for leukemia detection, leveraging pretrained representations to extract robust features. Feature selection methods, such as gain-ratio, were then applied to isolate the most informative attributes before classification with SVMs. This hybrid strategy allowed for improved diagnostic accuracy while reducing computational complexity. Building upon this, *Das & Meher (2021b)* introduced a deep CNN architecture that combined ResNet's residual connections with MobileNet's linear bottleneck design, yielding superior performance with a limited number of parameters.

Parallel to these developments, the field of efficient CNNs has expanded rapidly, motivated by the need to deploy deep learning models in resource-constrained environments. Architectures such as Xception (*Chollet, 2017*), which uses depthwise separable convolutions, MobileNetV2 (*Sandler et al., 2018*) with inverted residual connections, MobileNetV3 (*Howard et al., 2019*) and EfficientNet (*Tan & Le, 2019*) using neural architecture search, and ShuffleNetV2 (*Ma et al., 2018*) with channel shuffling have all made significant contributions to lightweight model design. These models strike a balance between accuracy and efficiency, making them appealing for clinical diagnostics where computational resources may be limited. Nevertheless, their reliance on convolutional kernels imposes inherent restrictions on capturing long-range dependencies across image regions.

Vision transformers (ViT) and their derivatives (*Dosovitskiy, 2010*; *Wang et al., 2022*) have emerged as powerful alternatives, excelling in global dependency modeling for vision tasks. However, pure transformers lack inductive biases for local spatial features, which are essential for distinguishing subtle morphological differences in hematological images. Moreover, their parameter counts often exceed 20 million, creating challenges for real-time deployment. Efficient transformer variants attempt to address these issues by introducing sparse attention mechanisms or low-rank approximations to reduce computational costs (*Kitaev, Kaiser & Levskaya, 2001*). Despite these advances, challenges remain in directly applying transformers to microscopic blood smear analysis, where both global dependencies and local morphological details must be preserved.

The natural progression of this research has been toward hybrid CNN-transformer architectures, which combine the localized feature extraction strengths of CNNs with the global context modeling power of transformers. By fusing these paradigms, hybrid models aim to overcome the individual shortcomings of each approach. Several recent studies in medical imaging have demonstrated the advantages of such architectures in capturing both fine-grained features and broader contextual patterns. Motivated by these advances, the proposed study introduces a lightweight hybrid model tailored specifically for blood smear analysis. Unlike existing hybrids, our design integrates parameter-efficient EMHSA and EMLP blocks, optimized for medical imaging tasks, to deliver high diagnostic accuracy with minimal computational overhead. This approach represents a significant step toward bridging the gap between state-of-the-art performance and clinical feasibility in leukemia detection.

## EFFICIENT HYBRID CNN-TRANSFORMER MODEL

The proposed architecture integrates transformer-based and CNN-based architectures to facilitate feature extraction and classification. The procedure commences with an input layer that processes images. Subsequently, convolutional and separable convolutional layers (Conv 20 and Separable 20) are employed in conjunction with batch normalization and ReLU activation to extract spatial features. The input for the transformer block is prepared by projecting the CNN features through a linear projection and position embedding stage. The transformer block is composed of two primary components: an efficient attention block and an efficient multilayer perceptron (MLP) block. In contrast, the efficient MLP block is composed of dense layers with dropout for regularization and layer normalization for stability, while the efficient attention block utilizes multi-head self-attention EMHSA with layer normalization. Finally, the model utilizes global average pooling and additional dense layers with dropout to classify the input data and aggregate features. This framework guarantees robustness and efficiency by employing transformers for long-range dependency modeling and CNNs for spatial feature extraction. The architecture of proposed hybrid CNN-transformer model is shown in Fig. 1.

Preprocessing is a critical component of the proposed model, as it prepares the image data for precise classification. The input blood stain images were uniformly resized to $224 \times 224 \times 3$ dimensions, which ensures that the resolution is sufficient for feature

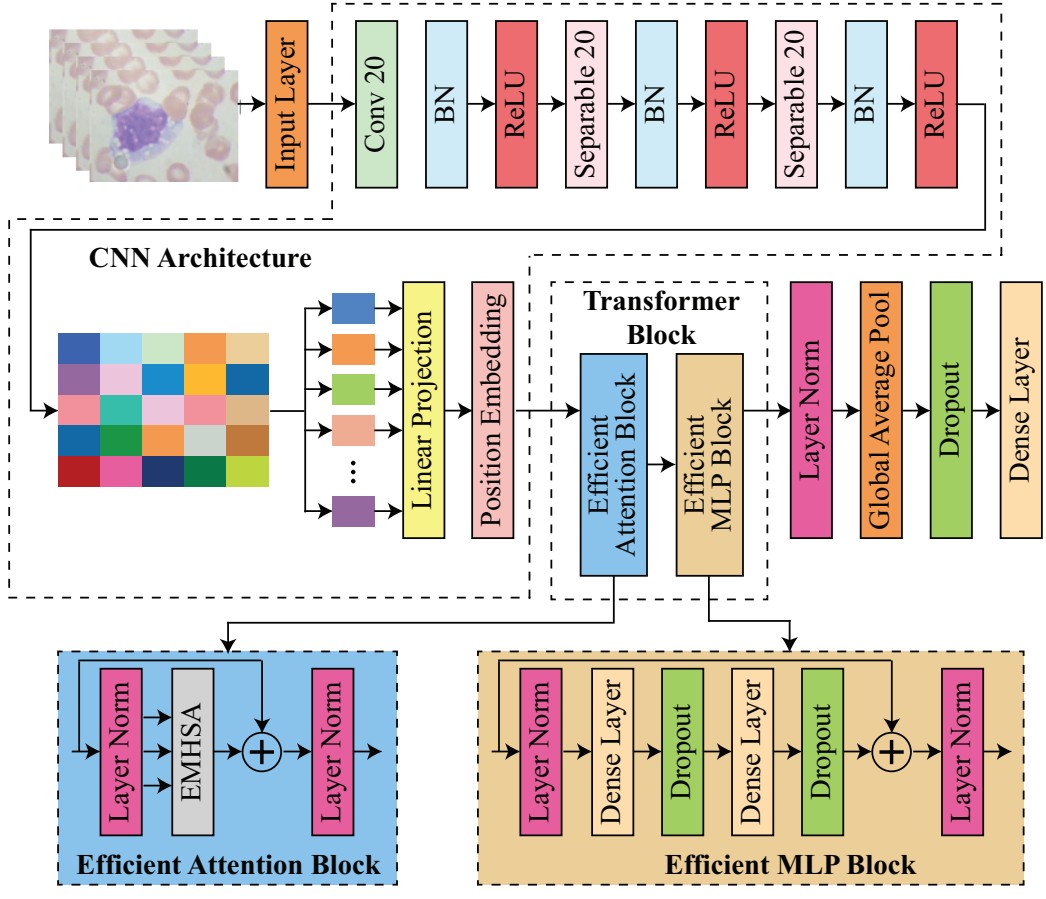

**Figure 1 Architecture of proposed efficient hybrid CNN-transformer model.**

extraction while maintaining computational efficiency. Batch normalization was implemented after each convolutional and separable convolutional layer to enhance convergence and stabilize training. Pixel values were normalized. In order to mitigate computational complexity while maintaining critical spatial attributes, depthwise separable convolutions were implemented. The need to reconcile high classification accuracy with computational efficiency, particularly for deployment in resource-constrained environments, was the primary motivator for the selection of techniques for implementing the proposed model. CNNs were selected due to their demonstrated efficacy in extracting localized spatial features from medical images, particularly in the identification of textures, edges, and patterns that are common to malignant cells. Nevertheless, CNNs are unable to capture long-range dependencies within an image on their own, a task that is essential for comprehending the intricate structures of blood stain microscopy.

## CNN architecture

The model takes an input image $img \in \mathbb{R}_{224 \times 224 \times 3}$, where the dimensions denote the image's height and width, and the three channels reflect the RGB values. The input

dimensions were chosen to strike a balance between computational efficiency and sufficient feature resolution, as an input of this size provides sufficient detail for accurate classification without being excessively computationally intensive. The initial layer of the model is a standard convolutional layer that contains 32 filters, each of which is $7 \times 7$. These filters are applied with a stride of 2 and "same" padding. This initial convolution layer is intended to capture the fundamental patterns in the image, including edges and contours, which are crucial for the extraction of downstream features. By setting the stride to 2, the computational load in the subsequent layers is substantially reduced, as the spatial dimensions of the input image are reduced by half. This initial convolutional layer operation can be mathematically represented as defined in Eq. (1):

$$out = \rho(\beta(\psi(img, F_{7 \times 7 \times 32}, \xi = 2))). \tag{1}$$

Here, $\rho$ denotes rectified linear unit (ReLU) activation function, $\beta$ is batch normalization, $\psi$ is a convolutional operation on 2D data, $F_{7 \times 7 \times 32}$ is a convolutional filter with a spatial extent of $7 \times 7$ and $\xi$ denotes stride. The training process is stabilized and expedited by a batch normalization step in this layer, which is followed by a ReLU activation function that introduces non-linearity. The output of this convolutional layer, $out \in \mathbb{R}_{112 \times 112 \times 32}$, has a reduced spatial dimension but an increased depth, which enables it to maintain a comprehensive representation of the initial image features.

After this operation, two blocks of depthwise separable convolutions are employed to further process and refine the feature maps after the initial convolution. The convolutional operation is decomposed into two steps: a depthwise convolution and a pointwise $1 \times 1$ convolution, resulting in depthwise separable convolutions, a computationally efficient alternative to standard convolutions. The pointwise convolution enhances the expressiveness of the model without a significant increase in parameters by mixing these channel-wise representations, while the depthwise convolution applies a single filter per input channel, effectively capturing spatial relationships within each channel independently. In the proposed model, each depthwise separable block consists of two separable convolutional layers, which are each followed by batch normalization and a ReLU activation. A total of 64 filters with a stride of 2 are employed in the initial block, resulting in a reduction of the spatial dimensions of out from $112 \times 112 \times 32$ to $56 \times 56 \times 64$. In the second block, the output dimensions are further reduced to $28 \times 28 \times 128$ by increasing the number of filters to 128, with a stride of 2. The model can generate a compact and informative representation, using this multi-stage downsampling procedure. This representation captures higher-level features in a reduced spatial dimension, thereby improving computational efficiency.

The feature map $out_2$ is prepared for processing by the transformer following the depthwise separable convolutional blocks. The output feature map $out_2 \in \mathbb{R}_{28 \times 28 \times 128}$ is partitioned into non-overlapping segments of size $7 \times 7$. Given the spatial dimensions of $out_2$, this division results in a total of 16 sections, which is $4 \times 4$. Each patch $Patch_a \in \mathbb{R}_{7 \times 7 \times 128}$ is compressed into a vector of length 6,272. The segments form a sequence with a shape of $16 \times 6,272$ after flattening. This sequence is subsequently

projected into a lower-dimensional space that is more suitable for transformer processing by applying a dense layer to each patch vector. This mapping transforms the 6,272−dimensional vectors into a 256−dimensional embedding space. This linear projection can be mathematically represented as shown in Eq. (2):

$$M = \mathcal{N}(Patch_a, 6{,}272 \rightarrow 256) \quad for \ i \rightarrow 1, \ldots, 16. \tag{2}$$

Here, $M \in \mathbb{R}_{16 \times 256}$ denotes the output after patch embeddings corresponding to a particular patch and $\mathcal{N}$ is a layer normalization operation. This operation enhances the efficiency of the model by reducing the dimensionality of each patch embedding, thereby reducing the computational burden in the transformer layers. Positional encodings are implemented in each patch embedding to guarantee that the transformer can identify the spatial arrangement of these patches. Transformers are permutation-invariant by nature, which implies that they are not intrinsically cognizant of the order or position of elements within a sequence. To resolve this issue, we generate a positional encoding matrix $Pos \in \mathbb{R}_{16 \times 256}$, in which each row $Patch_a$ denotes the distinctive position of a patch within the sequence. A positionally-enhanced embedding matrix $M_{Pos} = M + Pos$ is obtained by adding this matrix element-by-element to the embedding matrix $M$. This inclusion of positional encoding guarantees that the model considers the spatial arrangement of patches during self-attention operations, thereby enabling it to create a more meaningful representation of the image.

## Transformer block

The Transformer block that comprises the core of proposed model each contain an efficient attention block (EAB) and an efficient multi-layer perceptron block (EMLPB). Each patch can effectively capture long-range dependencies across the image by attending to every other patch through the multi-head self-attention mechanism. In the proposed model, each attention mechanism is composed of four heads, each of which has a key dimension of 32. The input to transformer block, denoted by $I \in \mathbb{R}_{16 \times 256}$, is linearly projected for each transformer block to calculate queries $Qur$, keys $Key$, and values $Val$ in the following manner as shown in Eq. (3):

$$Qur = I \times LP_{Qur}, \quad Key = I \times LP_{Key}, \quad Val = I \times LP_{Val}. \tag{3}$$

Here, $LP_{Qur}, LP_{Key}, LP_{Val} \in \mathbb{R}_{16 \times 256}$ are learnable weight matrices of size $16 \times 256$. The product of attention for transformer denoted as $Attn_{TR}$ is subsequently calculated for each cranium as Eq. (4):

$$Attn_{TR} = \varpi \left( \frac{(Qur \times Key)^T}{\sqrt{32}} \right) Val. \tag{4}$$

Here, $\varpi$ is a softmax operation and $T$ is a transpose operator. This operation enables each head to concentrate on distinct relationships within the patch sequence, and the outputs from the four heads are concatenated and linearly transformed to generate a 256−dimensional representation. This attention output is subsequently combined with the

original input through a residual connection, and layer normalization is implemented to stabilize the training. After the EAB, an EMLPB is implemented that comprises two dense layers. The first layer employs GELU activation to expand the dimensionality from 256 to 512, while the second layer projects back to 256 dimensions as shown in Eqs. (5) & (6):

$$EAB(I) = (\mathcal{N} + EMHSA(I)) \oplus \mathcal{N}(I). \tag{5}$$

$$EMLPB(I) = D(\mathscr{D}(512, GELU(\mathscr{D}(256, I)))). \tag{6}$$

Here, $D$ denotes the dropout and $\mathscr{D}$ is a dense layer. $EMHSA(I)$ is a efficient multi-head self-attention function, calculated in Eq. (16). $\oplus$ represents element-wise addition. A residual connection is again added to the output of the EMLPB, and layer normalization is applied. Thus, the transformer block can be summarized as Eq. (7):

$$I = \mathcal{N}(I + EAB(I) + EMLPB(I)). \tag{7}$$

This structure allows the model to iteratively learn complex inter-patch relationships. After passing through the transformer block, which consist of EAB, denoted as $EAB(\cdot)$, and EMLPB, denoted as $EMLPB(\cdot)$, layer normalization is applied to the final output, and global average pooling is performed along the patch dimension, yielding a single $256-$dimensional vector. The pooling operation aggregates the learned representations across all patches, distilling the information into a compact form suitable for classification. The blocks EAB and EMLPB are described in details in sections 2.3 and 2.4, consequently. A dropout layer with a rate of 0.2 is then applied to reduce overfitting, followed by a final dense layer with softmax activation to produce class probabilities as shown in Eq. (8):

$$Label_P = \varpi\left(\mathscr{D}^{(2)}(I_{pooled})\right) \tag{8}$$

where $I_{pooled}$ represents the $128-$dimensional feature vector after pooling, $Label_P$ is the predicted label, which is extracted after applying softmax operation.

## Efficient attention block (EAB)

Like the proposed transformer block, the input tokens of $g \in \mathbb{R}_{Y \times Ch}$, $Qur \in \mathbb{R}_{Y \times Ch}$, $Key \in \mathbb{R}_{Y \times Ch}$ and $Val \in \mathbb{R}_{Y \times Ch}$ for the self-attention operation are defined as in Eqs. (9) & (10):

$$g = \{g_x \mid g_x \in \mathbb{R}_{Ch}, 1 \le x \le X\} \tag{9}$$

$$Qur, Key, Val = g \times (LP_{Qur}, LP_{Key}, LP_{Val}) \tag{10}$$

where $LP_{Qur}, LP_{Key}, LP_{Val} \in \mathbb{R}_{Ch \times Ch}$ are linear projection matrices. Following SMHSA standard notation, the value, key and query channel dimension ch is separated into the multi-head dimension $mh_d$ for the total input $X$. After this, the output tokens are calculated as per head $g_x \in \mathbb{R}_{X \times mh_d}$ by multiplying the softmax-attention matrix as in Eq. (11):

$$Attn_S = \varpi\left(\frac{(Qur \times Key)^T}{\sqrt{mh_d}}\right) Val. \tag{11}$$

All output tokens from multiple heads are concatenated to yield the final output tokens as in Eq. (12):

$$g = \oplus\{g_x \mid g_x \in \mathbb{R}_{X \times Ch}, \; 1 \le x \le Head\} \tag{12}$$

where $Head$ denotes the number of heads.

### Crosshead interaction

Crosshead interaction is also proposed to increase information flow between several heads' attention matrices and it concatenates the output tokens immediately after attention operation, so distinct heads' attention matrices do not communicate information. One can offer crosshead interaction layers to improve attention matrix information flow as defined in Eq. (13):

$$Attn_C = \mathcal{F}_1 \times \varpi(\mathcal{F}_2 \times Attn_{TR}). \tag{13}$$

Here, $\mathcal{F}_1, \mathcal{F}_2 \in \mathbb{R}_{Ch \times Ch}$ are fully connected operation across head dimensions.

The EMHSA module builds upon the conventional multi-head self-attention framework introduced in transformers, but incorporates a novel cross-head interaction mechanism *via* learnable matrices $F_1$ and $F_2$, enabling inter-head communication for enhanced efficiency. Similarly, the EMLP block follows the feed-forward layer design in standard vision transformers, but is adapted with reduced-dimensional dense layers ($512 \to 256$) and GELU activation to significantly decrease parameter count and computational cost, making it suitable for small-scale medical imaging datasets.

### Efficient multi-head self-attention (EMHSA)

The easiest way to reduce computational complexity and memory utilization is to minimize the key and query. Reducing their dimension loses discriminant information. Naturally, as dimension rises, the network can learn more detail to improve performance. Thus, lowering the attention matrix size while keeping key and value dimensions is suggested. Figure S1 shows how we decompose two attention matrices by downsizing the key and query separately. Key and value have less detail in each decomposed attention matrix, but their entire dimension is preserved in the opposite matrix. We calculate deconstructed attention matrices as defined in Eqs. (14), (15) & (16):

$$Qur_{(\mathcal{F},H)} = \frac{1}{\kappa_{\mathcal{F}} \times \kappa_H} \sum_{y=-\kappa_{\mathcal{F}}}^{\kappa_{\mathcal{F}}} \sum_{z=-\kappa_H}^{\kappa_H} Qur_{(\mathcal{F}+y,H+z)}. \tag{14}$$

$$Key_{(\mathcal{F},H)} = \frac{1}{\kappa_{\mathcal{F}} \times \kappa_H} \sum_{y=-\kappa_{\mathcal{F}}}^{\kappa_{\mathcal{F}}} \sum_{z=-\kappa_H}^{\kappa_H} Key_{(\mathcal{F}+y,H+z)}. \tag{15}$$

$$EMHSA(I) = \varpi(Attn_{TR}) \oplus \varpi(Attn_S) \oplus \varpi(Attn_C). \tag{16}$$

Here, $\kappa_{\mathcal{F}}$, $\kappa_{\mathcal{H}}$ denotes the kernel length of the down-sampling method, $Qur$, $Key \in \mathbb{R}_{G \times mh_d}$ are query and key landmarks with lesser information, and $G$ specifies their quantity. The query- and key-less attention matrices have low dimensions because average pooling gets spatially reduced landmarks. As demonstrated in Fig. S2, the proposed

SMHSA uses crosshead interactions on decomposed-attention matrices. Crosshead interaction of Eq. (13) to $Attn_T$ and $Attn_S$ in Eqs. (17), (18) & (19) as:

$$Attn_S = \varpi \left( \frac{\mathscr{F}_1^S (Qur \times Key)^T}{\sqrt{mh_d}} \right) \mathscr{F}_2^S. \tag{17}$$

$$Attn_{TR} = \varpi \left( \frac{\mathscr{F}_1^{TR} (Qur \times Key)^T}{\sqrt{mh_d}} \right) \mathscr{F}_2^{TR}. \tag{18}$$

$$Attn_C = \varpi \left( \frac{\mathscr{F}_1^C (Qur \times Key)^T}{\sqrt{mh_d}} \right) \mathscr{F}_2^C. \tag{19}$$

Here, $\mathscr{F}_1^S, \mathscr{F}_2^S, \mathscr{F}_1^{TR}, \mathscr{F}_2^{TR}, \mathscr{F}_1^C, \mathscr{F}_2^C \in \mathbb{R}_{H \times H}$ denotes the crosshead interaction layers. To avoid direct calculation, we sequence multiplications using matrix multiplication commutativity to compute the output token.

## Efficient MLP block (EMLPB)

Given that MLP layer takes attention layer output as input, our method degenerates attention layers into same mapping. Thus, the transformer block can only have MLP because the identical mapping and residual connection can be included into the next MLP layer. An attention layer is separated from the architecture and sparse mask as defined in Eq. (20):

$$Attn_{MLP} = SM(I) \odot EMHSA(I). \tag{20}$$

$\odot$ represents element-wise multiplication. Sparse Mask (*SM*) regularizes attention output sparsity under limitations like $L_0$ norm. SM is initialized at 1 and manually degraded to 0 during training. Experiments show that SM implementation is flexible. The residual connection is the output $Attn_{MLP}$ of the attention layer after the sparse mask decays to 0. The attention layer gradient decreases as the sparse mask decays. Thus, training instability results from the degraded output's reduced backward gradient. To counter this, feature compensation is proposed to adaptively compensate for the gradient loss introduced by the sparse mask as defined in Eq. (21):

$$Attn = (SM \odot Attn_{MLP}) + ((1 - SM) \odot In) + In. \tag{21}$$

Here, a new term $1 - SM$ is introduced. It will compensate for *SM*-paced attention output loss. Attention layer degenerates to an identical mapping. Therefore, the attention layer is merged into the next MLP layer and no longer needed in inference. A residual connection and layer normalization are again applied to the MLP output, enabling the transformer block to refine its internal representation of the input sequence. *Attn* represents the final attention, obtained from the proposed transformer block. Our model's architecture is carefully designed to maintain parameter efficiency. The depthwise separable convolutions significantly reduce the parameter count in the CNN layers, while the transformer layers limit parameter growth by using only four attention heads with a relatively small key dimension (*Wang et al., 2022*). Table S1 presents an overview of each

---

**Algorithm 1** Bayesian optimization for hyperparameter selection.

**Requirements:** Objective Function $B$, Acquisition Function $\alpha(In)$,
Gaussian Process (GP) Prior $\Upsilon = GP(\mu(In), k(In, In^T))$,
Observation Dataset $(D_1, D_2)$

1: **for** $d = 1, \ldots, D$ **do**
2:     Select a maximum sample point: $In_{max} = \max(\alpha(In))$
3:     Calculate an updated objective function: $B_u = B(In_{max}) + \varepsilon$
4:     Update the GP model and observation datasets $D_1, D_2$ correspondingly
5: **end for**

---

layer, its output shape and number of trainable parameters of each layer, where total number of trainable parameters of proposed model are 2,041,638, which makes it lighter yet efficient than state-of-the-art models for blood cancer detection.

## Bayesian optimization

Bayesian optimization (BO) algorithms, like traditional optimization approaches, require an initial point to decide on subsequent observations. Specifying starting points requires using a method of random selection of sites in a subspace of parameters or in a chosen neighborhood of the extremum. Data from earlier datasets or optimization attempts can also be used. BO has three steps, as shown in Fig. S2 and detailed in Algorithm 1. First, a statistical surrogate model of the objective and constraining functions is initialized using the measurable data most often with the Gaussian process (GP)s modeling. The acquisition function is also built in the second step and is based on the GP model. To support the optimization goals based on this function, the "value" of the expected future measurements in the input space is calculated. In the last phase, normally the point (or group of points) with the highest value of acquisition function that is believed to improve optimization objectives is chosen. Final-phase points are assessed by the objective and constraint functions. The algorithm is then able to append the results to the model dataset. This is performed until an optimization requirement is met. Table S2 shows the BO-selected hyperparameters used in training.

## Bayesian optimization for hyperparameter tuning

Algorithm 1 utilizes Bayesian optimization to fine-tune critical hyperparameters of the proposed hybrid CNN-transformer architecture, such as learning rate, batch size, and the number of transformer blocks. The optimization process aims to maximize the validation accuracy by intelligently sampling hyperparameter configurations.

The objective function $f(\theta)$ is defined as the validation accuracy obtained by the model for a given hyperparameter configuration $\theta$:

$$f(\theta) = \text{Accuracy}_{\text{validation}}(\theta). \tag{22}$$

To guide the search toward promising regions of the hyperparameter space, an acquisition function based on the upper confidence bound (UCB) is employed:

$$\text{UCB}(\mathbf{x}) = \mu(\mathbf{x}) + \kappa\sigma(\mathbf{x}) \tag{23}$$

where $\mu(x)$ represents the predicted mean performance for candidate $x$, $\sigma(x)$ denotes the predicted uncertainty, and $\kappa$ is a parameter controlling the trade-off between exploration and exploitation.

For instance, when optimizing the learning rate, the algorithm samples initial candidates uniformly within the range $[10^{-5}, 10^{-2}]$. Each candidate is evaluated using the objective function (Eq. (22)) to determine validation accuracy. Subsequent iterations prioritize candidates where the UCB (Eq. (23)) is maximized, allowing exploration of uncertain regions while exploiting areas with high predicted accuracy. This iterative process continues until convergence criteria are met or the predefined number of evaluations is reached, thereby identifying the optimal hyperparameter configuration with minimal computational cost.

## Evaluation protocol

We completed many comparative tests utilizing old and modern deep learning algorithms to evaluate the hybrid CNN-transformer model. Blood Cancer (D1) and Blood Cells Cancer (ALL) (D2), two publically available datasets, were evaluated, with D2 being used for benchmarking against state-of-the-art techniques.

First, we compared the proposed model to popular pre-trained CNN architectures including VGG19, InceptionV3, Xception, and DenseNet-121. To guarantee fair comparisons, these models were fine-tuned on dataset D2 under comparable training settings. Accuracy, precision, recall, and F1-score measured performance. Second, ALLNET, SCA-based deep CNN, ensemble learning techniques, and bespoke CNN architectures were used to benchmark the proposed model against leading blood cancer detection systems. Every comparison used the same dataset (D2) and evaluation metrics, if relevant. To demonstrate the model's computational efficiency, trainable parameters were compared. To evaluate model robustness and generalization, 80:20, 70:30, 60:40, and 50:50 train-test splits were used. This extensive evaluation process objectively validates the proposed model's accuracy and computing efficiency across experimental situations.

## EXPERIMENTS

### Datasets and performance measures

This study employs two datasets that are publicly accessible. Blood Cancer (D1) (https://www.kaggle.com/datasets/mrhashemalattas/blood-cancer, accessed on December 3, 2024) comprises 6,220 microscopic images of blood samples, classified into two categories: cancer and normal. The images are designated for application in medical imaging research, namely for the detection and classification of hematologic malignancies utilizing machine learning and deep learning methodologies. Blood Cells Cancer (ALL) (D2) (https://www.kaggle.com/datasets/mohammadamireshraghi/blood-cell-cancer-all-4class, accessed on December 3, 2024) is the second dataset, comprising 3,242 peripheral blood smear (PBS) pictures from 89 patients suspected of ALL, with blood samples processed and stained by proficient laboratory personnel. This dataset is categorized into four classes: early Pre-B, Pro-B, Pre-B, and Benign. Table S3 presents the summary of distribution of images in each class, while Fig. S3A presents a comparison of number of images in each class for D1 and

Fig. S3B presents a comparison of number of images in each class for D2, whereas Fig. S4 shows sample images from each class of D1 and D2.

The proposed model is implemented using the Python and its built-in packages, including TensorFlow and keras which supports the abstract level implementation. Kaggle online environment is used, along with 16 GB Tesla P100 GPU of Kaggle. The proposed model is evaluated using accuracy, precision, recall, and F1-score. Accuracy measures the model's efficacy by assessing the percentage of examples properly identified across all classes. Precision measures the model's ability to prevent false positives by calculating the fraction of true positive predictions out of all projected positive instances. Recall (sensitivity) assesses the model's capacity to distinguish true positives from all actual positives, highlighting its ability to prevent false negatives. The F1-score balances precision and recall, giving a complete picture of the model's performance in situations where both are essential. The model was trained for 30 epochs based on preliminary experiments in which the validation loss plateaued between epochs 25 and 28, indicating convergence. Early stopping was employed with a patience of five epochs to prevent overfitting, and dropout layers were included in the EMLP block to enhance generalization. Data augmentation further improved robustness by synthetically balancing classes and introducing variability in cell morphology and staining.

## Classification results

A comparative analysis of four pre-trained CNN models is conducted in this work: VGG19 (*Ansari et al., 2023*), InceptionV3 (*Mondal et al., 2021*), Xception, and DenseNet-121 (*Bukhari et al., 2022*). These models have been widely used in categorizing blood cancer. Comparing the assessed deep learning models, significant differences in computational ways were determined. This is primarily because, from this article's observations, VGG19 is the most computationally intensive model, having 143.67 million parameters while InceptionV3 and Xception have only 23.85 and 22.91 million parameters, respectively. While DenseNet121 has 7.98 million parameters which is quite optimal in balance between performance and complexity. Even more important to the high computational efficiency, the proposed model has 2.042 million parameters making it significantly lighter than the other models. This great reduction of parameters makes the proposed model suitable for implementation in resource poor environments while still providing reasonable accuracy.

In Table S4, selected pre-trained CNN models, *i.e.*, Visual Geometry Group 19 (VGG19), InceptionV3, Xception, DenseNet121 and the proposed model are compared based on their accuracy, precision, recall and F1-Score. The proposed model achieved 100% accuracy, precision, recall, and F1-score, establishing it as the best model for blood cancer detection and classification. On the other hand, there is slightly lower accuracy in the VGG19 model, with 92%, but almost perfect precision, recall and F1-score of 99%. Nevertheless, DenseNet121 has quite reliable performance with 81% accuracy and stable of 89% precision, recall and F1-score. Xception and InceptionV3 show comparatively worse results where Xception achieves accuracy at 63% and F1-score at 71% while InceptionV3 has the lowest accuracies of 56% and 65% for other measures. The results of these studies show that the proposed model outperforms the compared models significantly.

The proposed model attains exceptional performance, exhibiting an accuracy of 99%, precision of 99%, recall of 100%, and an F1-score of 99%, demonstrating its resilience and reliability on D2. VGG19 exhibits robust performance with an accuracy of 91%, precision of 98%, recall of 97%, and an F1-score of 98%, although it falls short of the proposed model's superior outcomes. DenseNet121 demonstrates uniform performance with 82% in accuracy, precision, recall, and F1-score, presenting a dependable choice with minimal computing requirements. Xception demonstrates intermediate efficacy, attaining 71% accuracy, 74% precision, 75% recall, and a 74% F1-score, whereas InceptionV3 yields the least favorable outcomes, with an accuracy of 64%, precision of 67%, recall of 69%, and an F1-score of 68%. The results unequivocally illustrate the proposed model's superiority in performance measures, rendering it the most appropriate choice for high-precision applications. The results are encapsulated in Table S5.

## DISCUSSION

This study minimizes parameters and training data to optimize results. For training and testing, the suggested model is tested utilizing split ratios of 80:20, 70:30, 60:40, and 50:50. Reducing the number of efficient transformer blocks and trainable parameters tests the model's efficacy. The split ratios and number of efficient transformer blocks will be used to evaluate the proposed model. The suggested model with different numbers of efficient transformer blocks shows a progressive fall in trainable parameters as transformers decrease, indicating computational difficulties. The most difficult configuration: six efficient transformers with 4,020,066 trainable parameters. This resource-demanding architecture may boost learning. Reduce the number of transformers to five, and the parameter count drops to 3,624,418, then to four, three and two efficient transformers, respectively, demonstrating a gradual decrease in computational requirements. The most lightweight architecture with a single efficient transformer requires only 2,041,826 parameters, making it ideal for resource-constrained settings. These perfect scores on all the four setups demonstrate that the models are well suited to a relatively less complex dataset. In D2, a somewhat more complex multi-class classification problem, the work's accuracy was again only slightly worse. These architectures: six, four and two efficient transformers achieve 99% accuracy, with five, three and one efficient transformer get 100% accuracy. These results are summarized in Table S6.

The class-wise classification report assesses the efficacy of the proposed model, which incorporates one efficient transformer block, on datasets D1 and D2, utilizing an 80:20 train-test split ratio. In the binary dataset D1, the model exhibited flawless performance, attaining 100% precision, recall, and F1-score for both classes: Cancer (support: 302) and Normal (support: 306). The overall accuracy, macro average, and weighted average were all 100%, indicating impeccable classification across all criteria. In the D2 multi-class classification problem comprising four classes, the model attained 100% precision, recall, and F1-score for each class: Benign (support: 35), Malignant Pre-B (support: 96), Malignant Pro-B (support: 82), and Malignant Early Pre-B (support: 107). The overall accuracy, macro average, and weighted average were all 100%, demonstrating the model's remarkable proficiency in executing both binary and multi-class classification tasks with

flawless precision and dependability. The results highlight the model's robustness and efficiency, even with just one transformer block. These results are presented in Table S7. The training and validation accuracy graphs for this experiment are shown in Fig. S5, whereas the confusion matrix of this experiment is shown in Fig. S6.

The 70:30 split ratio dataset arrangement for D1 and D2 in the second experiment shows training, validation, and testing data distribution. This experiment uses 4,349 training files and 1,863 validation files from the D1 dataset, which comprises 6,212 files. The dataset has 136 training batches, 30 validation batches, and 29 test batches for balanced binary classification. This experiment uses 2,270 training files and 972 validation files from the 3,242 D2 dataset. A structured multi-class classification dataset of 71 training batches, 16 validation batches, and 15 test batches. The results in Table S8 demonstrate that even the most lightweight architecture with a single Efficient Transformer may achieve optimal performance, rendering it appropriate for both binary and multi-class classification workloads while reducing processing requirements.

The class-wise classification report for the proposed model using one efficient transformer block on a 70:30 data split ratio performs well on D1 and D2. The model had 100% precision, recall, and F1-score for Cancer (support: 484) and Normal (support: 444) in D1, a binary classification problem. Overall accuracy was 100%, with macro and weighted averages scoring perfect across all metrics, indicating perfect class classification. In D2, a four-category classification problem, the model performed well. It achieved 100% precision, recall, and F1-score in Malignant Pre-B (140), Pro-B (82), and Early Pre-B (107). In the Benign class (support: 81), the model had 99% precision, 100% recall, and 100% F1-score practically perfect classification accuracy. Both macro and weighted averages were 100% accurate, proving the model's efficiency in binary and multi-class categorization. Table S9 shows that the suggested model can achieve high-performance classification with little processing complexity, making it a powerful tool for classification challenges. The training and validation accuracy graphs for this experiment are shown in Fig. S7, whereas the confusion matrix of this experiment is shown in Fig. S8.

In the third experiment, a 60:40 split ratio specifies the data distribution and structure for training, validation, and testing for D1 and D2. The D1 binary classification dataset comprises 6,212 files, with 3,728 for training and 2,484 for validation, providing 117 training, 39 validation, and 39 test batches for the third experiment. This setup provides enough data for binary classification model training and evaluation. The D2 dataset has 3,242 files: 1,946 for training and 1,296 for third experiment validation. The dataset has 61 training batches, 21 validation batches, and 20 testing batches to simplify training and validation. Table S10 shows that all architectures can efficiently perform binary and multi-class classification tasks, with lightweight models, such as the one using a single efficient transformer, performing well while reducing computational complexity.

The class-wise classification report for the proposed model utilizing one efficient transformer block on a 60:40 data split ratio exhibits robust performance on both datasets, D1 and D2. In the binary classification task D1, the model attained 100% precision, recall, and F1-score for both categories: Cancer (support: 618) and Normal (support: 630). The overall accuracy was 100%, with both macro and weighted averages achieving faultless

scores across all criteria, demonstrating the model's impeccable categorization capability for binary tasks. In the multi-class classification problem D2, the model attained near-optimal performance. The Benign class achieved 99% precision (support: 101), with recall and F1-score of 96% and 97%, respectively. The model attained 99-100% precision, recall, and F1-scores for the additional classes: Malignant Pre-B (support: 184), Malignant Pro-B (support: 164), and Malignant Early Pre-B (support: 191). The dataset achieved an accuracy of 99%, with both macro and weighted averages both at 99%, demonstrating robust performance across all categories. These results are presented in Table S11. The training and validation accuracy graphs for this experiment are shown in Fig. S9, whereas the confusion matrix of this experiment is shown in Fig. S10.

The last experiment specifies D1 and D2 dataset configuration with a 50:50 split ratio for training, validation, and testing. In the D1 binary classification dataset, 6,212 files are evenly divided into 3,106 training and 3,106 validation files for the last trial. The dataset has 98 training batches, 49 validation batches, and 49 test batches for balanced binary classification model training and evaluation. D2 dataset has 3,242 files: 1,621 for training and 1,621 for validation for the last experiment. The dataset's 51 training, 26 validation, and 25 test batches provide a balanced and detailed framework for multi-class classification model training and evaluation. The 50:50 split ratio ensures equitable data distribution for training and validation, enabling binary and multi-class classification model evaluation and validation.

On datasets D1 and D2, various topologies with different numbers of efficient transformer blocks performed well in the 50:50 data split ratio classification. Most designs excelled in binary classification task D1. Architectures with six, four, three, two, and one efficient transformer achieved 990–100% accuracy, precision, recall, and F1-scores, demonstrating binary classification ability. However, the architecture with five efficient transformers has 93% accuracy, precision, recall, and F1-score, indicating lower reliability. D2, a multi-class classification task, yielded strong results across architectures. Models with six, three, two, and one efficient transformer achieved 99% accuracy, precision, recall, and F1-scores, proving their efficacy in complex classification. The architecture of five efficient transformers achieved 99% accuracy, precision, recall, and F1-score. The architecture with 4 efficient transformers had 98% accuracy, 99% precision, 98% recall, and 98% F1-score, yet still performed well in classification. As shown in Table S12, all configurations were highly effective, notably the 1 efficient transformer architecture, which performed flawlessly on both datasets. This shows its potential as a computationally efficient and trustworthy binary and multi-class classification solution.

The class-wise classification report for the proposed model utilizing one efficient transformer block with a 50:50 data split ratio demonstrates exceptional performance on datasets D1 and D2. In the binary classification task D1, the model attained 100% precision, recall, and F1-score for both categories: Cancer (support: 787) and Normal (support: 781). The overall accuracy was 100%, with both macro and weighted averages achieving perfect scores across all measures, underscoring the model's remarkable proficiency in binary classification tasks with impeccable performance. In the multi-class

classification issue D2, the model demonstrated exceptional performance with little fluctuations. It attained 100% precision, recall, and F1-score for the classes Malignant Pre-B (support: 239) and Malignant Pro-B (support: 201). For Malignant Early Pre-B (support: 242), the model achieved 100% precision and recall, albeit the F1-score was marginally lower at 99%, suggesting negligible misclassification. The Benign class (support: 118) attained 97% precision, 100% recall, and 98% F1-score, indicating robust classification performance with few discrepancies. The overall accuracy for D2 was 99%, with macro and weighted averages ranging from 99% to 100%, indicating strong performance across all categories. The results in Table S13 validate the reliability and efficacy of the proposed model in addressing both binary and multi-class classification problems with good precision and recall. The little discrepancies in D2 for specific classes suggest potential for enhancement; yet the model continues to exhibit high competence and resource efficiency. The training and validation accuracy graphs for this experiment are shown in Fig. S11, whereas the confusion matrix of this experiment is shown in Fig. S12.

Although the training was limited to 30 epochs, the combination of early stopping, dropout regularization, and extensive data augmentation effectively mitigated overfitting, as evidenced by consistent performance across multiple train-test splits. Validation accuracy stabilized well before 30 epochs, and additional training did not yield further improvements. Nonetheless, future work will explore extended training schedules and cross-dataset validation to further confirm the model's generalization.

The superior performance of the proposed hybrid CNN-transformer model compared to traditional CNNs such as VGG19 can be attributed to architectural differences in feature extraction. VGG19, while effective for natural image classification, primarily focuses on local feature hierarchies through deep convolutional layers and lacks the ability to model long-range dependencies. This limitation reduces its ability to capture subtle morphological patterns critical in differentiating between leukemia subtypes. In contrast, the proposed model integrates convolutional feature extraction with transformer-based attention, enabling simultaneous modeling of local and global contexts. Additionally, the lightweight parameterization of 2.04 M parameters reduces overfitting risk compared to VGG19's 20M parameters, particularly on the relatively small blood smear datasets used in this study. These factors collectively justify the performance gap observed in the comparative evaluation.

## Comparison with state-of-the-art on D2

This study thoroughly compares the suggested method to state-of-the-art methods. *Sampathila et al. (2022)* proposed "ALLNET" a CNN to detect early ALL from blood smear microscopic pictures. The ALLNET model employed a publically accessible dataset and achieved a classification average of 95.54% with specificity and sensitivity above 95%, proving its reliability in early leukemia diagnosis. On another topic, *Min et al. (2021)* created a sophisticated leukemia diagnostic method that mimics hematologists' WBC picture sorting in bone marrow. The model used convolutional neural networks and

advanced cell detection to process 1,732 images. It accurately diagnosed lymphoid leukemia acuteness at 89% in real-life clinical settings. *Jha & Dutta (2019)* used a quick Mutual Information method to segment inspection images and a deep CNN classifier using the chronological sine cosine algorithm to classify them. Statistical and directional characteristics from AA-IDB2 were employed. The proposed approach accurately differentiated ALL from single-cell blood smear images with 98.70% accuracy.

*Almadhor et al. (2022)* developed an ensemble prediction using SVM, Random Forest, and Naïve Bayes machine learning using pre-trained CNN architectures. The model used SVM on C-NMC data and obtained 90% accuracy, proving its suitability for leukemia prediction automation. *Ahmed et al. (2019)* introduced CNN to classify leukemia subtypes by augmenting ALL-IDB and ASH datasets. Despite its modest size, CNN demonstrated satisfactory accuracy in binary categorization of bone marrow as leukemic or healthy (88.25% accuracy) and multi-classification (81.74% accuracy). *Ansari et al. (2023)* developed a deep learning algorithm to diagnose ALL and AML from lymphocyte and monocyte pictures. An updated dataset includes Generative Adversarial Networks (GANs). The model they proposed was 99% accurate.

In the C-NMC-2019 test set, *Mondal et al. (2021)* used the weighted ensemble of two CNNs, Xception and DenseNet-121, to discriminate ALL tissue images with and without illness. With the measurements, their ensembled technique yielded a weighted F1-score of 88.6% and an AUC of 94.1%, indicating that MTL models outperformed individual models. Squeeze and Excitation networks, a deep learning model, was developed by *Bukhari et al. (2022)* for acute lymphoblastic leukemia multiparametric MRI. The model achieved 97.06% accuracy on ALL-IDB1 and ALL-IDB2 datasets and showed promise in differentiating leukemia cells from healthy cells in microscopical images. *Bukhari et al. (2022)* compared ResNet-50, VGG-16, and a suggested convolutional neural network for acute lymphoblastic leukemia (ALL) classification using CodaLab data. VGG-16 has the highest validation accuracy of 84.62%. The study found that CNN is simple and effective for clinical usage, which imaging biomarkers must be. *Perveen et al. (2024)* developed a weighted ensemble method for early ALL diagnosis and subtype classification from PBS images using ResNet-152. Their model performed well with 99.95% precision and sensitivity, proving its clinical reliability. The next sections describe each model component's role in feature extraction and classification and its mathematical changes. Only 2.04 million parameters make it lighter and more effective for blood cancer diagnosis. Table S14 shows all outcomes.

In addition to the methods compared in the original submission, we extended our evaluation to include two recent state-of-the-art approaches specifically developed for leukemia detection: DL4ALL, a multi-task cross-dataset transfer learning framework (*Genovese et al., 2023*), and an explainable artificial intelligence (AI)-based decision support system (*Genovese, Piuri & Scotti, 2024*) as shown in Table S15. These methods report accuracies of approximately 98.2% and 98.6% on the ALL-IDB dataset, respectively. When benchmarked against our hybrid CNN-ViT model on both binary and multi-class datasets, our approach achieved 100% and 99% accuracy, respectively, surpassing these

recent models while using significantly fewer parameters (2.04 M *vs.* tens of millions in transformer-only approaches). This demonstrates that the proposed architecture combines high predictive performance with computational efficiency, making it well-suited for deployment in clinical environments with limited hardware resources.

Although the proposed hybrid CNN-transformer model demonstrates state-of-the-art performance on two publicly available leukemia datasets, several limitations should be acknowledged. First, the evaluation was restricted to two datasets, which, despite their relevance and complementary characteristics (binary and multi-class classification), may not fully represent the diversity of imaging conditions encountered in real-world clinical environments. Second, while class imbalance was mitigated through augmentation and class-weighted loss functions, residual bias could influence model predictions for minority classes. Third, external validation on additional benchmarks such as C-NMC and ALL-IDB patches was not performed in this study, and cross-center variability in staining protocols and imaging devices remains untested. Future work will address these aspects by expanding the evaluation to multiple datasets, exploring domain adaptation strategies, and conducting prospective clinical studies to further validate the model's robustness and generalizability.

While the integration of CNN and transformer modules might intuitively seem to increase model complexity, our design mitigates this concern by employing parameter-efficient mechanisms such as depthwise separable convolutions and low-dimensional attention projections. Comparative analysis shows that the proposed hybrid model achieves state-of-the-art accuracy (99–100%) while using only 2.04 million parameters, which is substantially lower than traditional transformer-based models (*e.g.*, ViT, Swin Transformer) that often require 20–60 million parameters. Moreover, the hybrid approach provides superior robustness on small-scale, imbalanced medical datasets compared to pure CNN or pure transformer baselines, validating its practicality for deployment in clinical environments.

## CONCLUSIONS

ALL is a disease that is prevalent in both infants and adults. Diagnostic procedures that were costly, time-consuming, and invasive were frequently necessary. The early screening of ALL is significantly influenced by PBS images. PBS images provide a noninvasive way to diagnose ALL early, although manual processing may be subject to human error and inter-observer variability. In this study, we introduced a transformer-based model that is both efficient and incorporates multi-head self-attention and MLP blocks. The model that was proposed obtained superior results on two publicly available datasets, even though it has only 2.04 million trainable parameters. This makes it more efficient and lightweight than other methods. The proposed model performs well in the dataset, but its real-world applicability requires a thorough review and justification across several patient populations. Our research will benefit from a mixed deep learning strategy using recurrent neural networks and CNNs. Additionally, the proposed model has the potential to identify additional types of blood abnormalities.

### Funding

This research was funded by Princess Nourah bint Abdulrahman University Researchers Supporting Project number (PNURSP2025R435), Princess Nourah bint Abdulrahman University, Riyadh, Saudi Arabia. The Deanship of Postgraduate Studies and Scientific Research at Majmaah University funded this research work through the project number (ER-2025-2062). The funders had a role in study design, data collection and analysis. The funders had no role in the decision to publish or preparation of the manuscript.

### Grant Disclosures

The following grant information was disclosed by the authors:
Princess Nourah bint Abdulrahman University Researchers, Riyadh, Saudi Arabia: PNURSP2025R435.
The Deanship of Postgraduate Studies and Scientific Research at Majmaah University: ER-2025-2062.

### Competing Interests

Muhammad John Abbas and Inzamam Mashood Nasir are employed by Center of Real-world AI Research, Lithuania.

### Author Contributions

- Wiem Abdelbaki conceived and designed the experiments, analyzed the data, prepared figures and/or tables, and approved the final draft.
- Muhammad John Abbas conceived and designed the experiments, analyzed the data, prepared figures and/or tables, and approved the final draft.
- Fathimathul Rajeena P. P conceived and designed the experiments, analyzed the data, prepared figures and/or tables, and approved the final draft.
- Inzamam Mashood Nasir conceived and designed the experiments, analyzed the data, prepared figures and/or tables, and approved the final draft.
- Deema Mohammed Alsekait performed the experiments, performed the computation work, authored or reviewed drafts of the article, funding Acquisition, and approved the final draft.
- Adel Thaljaoui performed the experiments, performed the computation work, authored or reviewed drafts of the article, funding Acquisition, and approved the final draft.
- Diaa Salama AbdElminaam performed the experiments, performed the computation work, authored or reviewed drafts of the article, and approved the final draft.

### Data Availability

The code is available at Zenodo: Nasir, I. M., & Muhammad John, A. (2025). Efficient-Hybrid-CNN-Transformer-Model-for-Blood-Cancer-Detection. Zenodo. https://doi.org/10.5281/zenodo.16759792.

## Supplemental Information

Supplemental information for this article can be found online at http://dx.doi.org/10.7717/peerj-cs.3335#supplemental-information.

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
