# Peer review of "Efficient hybrid CNN-transformer model for accurate blood cancer detection"

_PeerJ Computer Science, doi:10.7717/peerj-cs.3335_

## Round 0.1 · original submission · Major Revisions

· Academic Editor

Major Revisions

The proposed hybrid CNN-ViT architecture is used to detect leukemia in white blood cells. The hybrid architecture is designed to leverage the spatial localization capabilities of the CNN and the attention module in ViT, while also being more lightweight than current approaches in the literature. Please highlight the contribution and follow the reviewers' requests.

Reviewer 1 ·

Basic reporting

-

Experimental design

-

Validity of the findings

-

Additional comments

1. Clear and unambiguous, professional English used throughout

2. Good introduction with recent related works. The authors are appreciated.

3. The abstract should be updated with pertinent statistics in addition to the number of evaluation parameters and prediction accuracy.

4. Each block, which was previously created and made public, is described in great depth. Provide as much detail as possible using pertinent mathematics and references.

5. Transformers are capable of anticipating everything. There is no rationale for combining CNN and Transformer, save for accuracy. In fact, it makes the model more complex and raises the system's computing cost.

6 There are no sample images given from each data set for reference

7. Data imbalance is there, which will lead to improper results

8. There is a mismatch in the number of input images and the confusion matrix values of images

9. The results are overfitting. We cannot come to a conclusion by running just 30 epochs.

10. Why do the authors use six effective transformers when they only need one to achieve 100% accuracy?

11. Recall and other performance metrics are not strictly mentioned as REC …

Cite this review as

·

Basic reporting

The authors propose a hybrid CNN-ViT architecture for the purpose of detecting leukemia in white blood cells. The hybrid architecture is designed to take advantage of the spatial localization abilities of the CNN and the attention module in ViT, at the same time being more lightweight than current approaches in the literature.

I believe one of the major flaws of the paper lies in the unclear definition of the novelty: it is unclear whether the efficient modules described in the paper (efficient multi-head self-attention, Efficient MLP Block) are novel or are taken from previous works.

Experimental design

The second major flaw lies in the experimental evaluation: the models chosen as a comparison in Table 4 do not reflect the state of the art for leukemia detection. I would suggest that the authors compare their approach with more recent methods, designed for leukemia detection. The authors did a partial evaluation in Table 14, though only on D2 and without the most recent approaches (2024-2025). E.g.,

A. Genovese, V. Piuri, and F. Scotti, "A Decision Support System for Acute Lymphoblastic Leukemia Detection based on Explainable Artificial Intelligence", in Image and Vision Computing, vol. 151, no. 105298, November, 2024. ISSN: 0262-8856.

A. Genovese, V. Piuri, K. N. Plataniotis, and F. Scotti, "DL4ALL: Multi-task cross-dataset transfer learning for Acute Lymphoblastic Leukemia detection", in IEEE Access, vol. 11, 2023, pp. 65222-65237. ISSN: 2169-3536.

Also, a cross-fold validation is necessary to produce more meaningful results.

It would also be interesting to see results on the C-NMC database and on ALL-IDB patches: https://ieeexplore.ieee.org/document/10193429

Validity of the findings

-

Additional comments

The novelty must be more clearly defined. At the current stage, it might appear as a mere application of an existing CNN-ViT hybrid method. Moreover, such a hybrid architecture has to be put in context with other hybrid-based architectures proposed in the literature.

I believe the experimental part could be shortened by removing the loss plots and the confusion matrices, and showing the most significant results (e.g., the comparison between different variants and the comparison with the literature).

·

Basic reporting

The paper presents a promising model, and it's a well-written paper.

Experimental design

The experimental design needs to be compared with other models, which is why it has not achieved the same accuracy as the proposed model.

Validity of the findings

The validity of the results is very accurate; therefore, it requires strong justification.

Additional comments

The paper presents a hybrid CNN-transformer model that obtained superior results on two publicly available datasets.

Most of the comments have been addressed, but the related work section is still missing. The author needs to add related work after the introduction and compare the proposed hybrid CNN-transformer model with other published CNN-transformer models. Also, they need to justify why it only applies to two datasets.

Please cite the equation in the text.

The requirements of algorithm one need to be explained with an example, with the objective function and the acquisition function.

It is very impressive that the proposed model achieved 100% accuracy, but the author may justify why other models, like VGG19, have less accuracy.

The discussion part can be added with work limitations.

---

## Round 0.2 · Major Revisions

· Academic Editor

Major Revisions

Please address all requests and comments of the reviewers thoroughly.

Reviewer 1 ·

Basic reporting

Good

Experimental design

Good

Validity of the findings

Good

Additional comments

The reviewer's comments were incorporated nicely

Cite this review as

·

Basic reporting

I believe the authors did a good job in addressing the comments by the reviewers. However the file I was sent for review didn't contain any figure or table. If this is part of the authors' idea to streamline the paper, it is too much.

Experimental design

The authors addressed the experimental issues.

Validity of the findings

The authors addressed the experimental issues.

Additional comments

I believe the authors did a good job in addressing the comments by the reviewers. However the file I was sent for review didn't contain any figure or table. If this is part of the authors' idea to streamline the paper, it is too much.

·

Basic reporting

Related work and comparison with other published materials are considered the backbone of the paper. Through related work, readers get insight into the authors’ work and other closely related materials.
I suggest adding a related work section as a separate section consisting of two or three pages to make the paper more valuable.

Experimental design

Experiments are sound

Validity of the findings

Findings are adequate

Additional comments

Additionally, please resubmit the cover letter with line numbers and page numbers so I can see exactly where your previous comments have been addressed.

Submit the same cover letter with previous comments, along with line and page numbers

Also, add a related work section.

---

## Round 0.3 · accepted · Accept

· Academic Editor

Accept

Thank you for your contribution to our journal.

·

Basic reporting

nothing to report

Experimental design

nothing to report

Validity of the findings

nothing to report

Additional comments

nothing to report